# Development and validation of an age-sex-ethnicity-specific metabolic syndrome score in the Chinese adults

Shujuan Yang [1,2,11], Bin Yu[2,3,11], Wanqi Yu[2], Shaoqing Dai [2,4], Chuanteng Feng[2,3], Ying Shao[5], Xing Zhao[1], Xiaoqing Li[6], Tianjing He[7] & Peng Jia [2,8,9,10]

Metabolic syndrome (MetS) is characterized by metabolic dysfunctions and could predict future risk for cardiovascular diseases (CVDs). However, the traditionally defined dichotomous MetS neither reflected MetS severity nor considered demographic variations. Here we develop a continuous, age-sex-ethnicity-specific MetS score based on continuous measures of the five metabolic dysfunctions (waist circumference [WC], triglycerides [TG], high-density lipoprotein cholesterol [HDL-C], mean arterial pressure [MAP], and fasting blood glucose [FBG]). We find that the weights of metabolic dysfunctions in the score vary across age-sex-ethnicity-specific subgroups, with higher weights for TG, HDL-C, and WC. Each unit increase in the score is associated with increased risks for hyperlipidemia, diabetes, and hypertension, and elevated levels of HbA1c, cholesterol, body mass index, and serum uric acid. The score shows high sensitivity and accuracy for detecting CVD-related risk factors and is validated in different geographical regions. Our study would advance early identification of CVD risks and, more broadly, preventive medicine and sustainable development goals.

Metabolic syndrome (MetS) is mainly characterized by a clustering of metabolic dysfunctions, including impaired glucose tolerance, dyslipidemia, hypertension, and central obesity[1]. These metabolic dysfunctions, also referred to as MetS components, are considered the interrelated risk factors of cardiovascular diseases (CVDs)[2,3], which have been a primary cause of death and disability worldwide. Therefore, MetS has been used as a biomarker that can predict the future risk for CVDs; for example, individuals with MetS may have a 2-fold risk of developing CVDs over the next 10 years than those without MetS[4], and 4.1 times higher risk of progressing to Type 2 diabetes[3].

According to the diagnosis criteria of Adult Treatment Panel III (ATP-III)[2], MetS is usually defined as the presence of three or more of the five common metabolic dysfunctions, i.e., elevated waist circumference (WC), elevated triglycerides (TG), reduced high-density lipoprotein cholesterol (HDL-C), elevated blood pressure (BP), and elevated fasting blood glucose (FBG). However, such a dichotomous definition for MetS, although helpful to identify certain populations at disproportionate future risk for CVDs[5,6], cannot reflect the severity of MetS, which has challenged its applications in tracking the progression of CVD risk and hence preventing CVDs. Previous studies have developed a continuous MetS score in western countries to determine the

[1]West China School of Public Health and West China Fourth Hospital, Sichuan University, Chengdu, China. [2]International Institute of Spatial Lifecourse Health (ISLE), Wuhan University, Wuhan, China. [3]Institute for Disaster Management and Reconstruction, Sichuan University, Chengdu, China. [4]Faculty of Geo-information Science and Earth Observation, University of Twente, Enschede, the Netherlands. [5]Yunnan Center for Disease Control and Prevention, Kunming, China. [6]Fujian Provincial Center for Disease Control and Prevention, Fuzhou, China. [7]Hubei Provincial Center for Disease Control and Prevention, Wuhan, China. [8]School of Resource and Environmental Sciences, Wuhan University, Wuhan, China. [9]Hubei Luojia Laboratory, Wuhan, China. [10]School of Public Health, Wuhan University, Wuhan, China. [11]These authors contributed equally: Shujuan Yang, Bin Yu. ✉e-mail: rekiny@126.com; jiapengff@hotmail.com

MetS severity, which was considered a compelling means of motivating patients that may feel empowered by a decrease in score[7,8]. Nevertheless, to the knowledge of the authors, there have not been such efforts in eastern countries where the components and progression of MetS may differ due to different genetic, demographic, socioeconomic, and lifestyle characteristics. This is particularly important in large eastern countries, such as China, which has a large population and has been facing a growing burden of CVDs[9]. It is therefore urged to develop a continuous MetS score on the basis of the Chinese population, to strengthen CVD prevention among about one-fifth of the world population.

In addition, as our understanding of MetS advances, some recent epidemiological studies have demonstrated different associations between MetS components and CVD risks across demographics, such as sex, age, and ethnicity[10–12]. Such variations have been completely hidden in the one-for-all MetS definition, which may in particular affect the accurate assessment of MetS severity in China, in which about one-tenth of the population are minorities. Therefore, the consideration of demographic variations in the MetS definition would be crucial for personalizing CVD prevention strategies.

To fill the two aforementioned gaps, we developed a continuous, age-sex-ethnicity-specific MetS score to assess the severity of MetS in China on the basis of the China Multi-Ethnic Cohort (CMEC), which is the largest multi-ethnic prospective cohort study in China, with about 44% of the participants being minorities[13]. To demonstrate the clinical usefulness of the MetS score for indicating the future risk for CVDs, we validated it in a prospective cohort study on the basis of the CMEC follow-up dataset. Also, to ensure the robustness of the MetS score for broader clinical use in all demographic subpopulations, especially in minorities, we further validated it in the three independently established cohorts of 134,403 residents in Yunnan, Hubei, and Fujian Province of China (Fig. 1). The findings of this study would provide a better understanding of the MetS severity and CVD risk in the Chinese. Also, an online MetS score calculator was developed and made available to the public, which can facilitate the use of the MetS score and this MetS severity assessment tool in clinical practice. The new knowledge and assessment tools resulting from this study are expected to improve individual-level CVD risk prediction and population-level CVD prevention and management in China and, more broadly, in other eastern countries.

## Results

### Development of the MetS score

Of the 77,639 participants at CMEC baseline, the mean age was $50.2 \pm 11.1$ years, with 39.0% (30,295) being male and 42.3% (32,857) being minorities (Table S1). The prevalence of MetS was 19.4% and varied across demographic subgroups, e.g., higher in those aged ≥60 than aged <60 (24.2% vs. 18.1%), in males than females (31.0% vs. 17.0%), and in minorities than Hans (19.7% vs. 19.1%) (Table 1).

Overall, TG had the highest factor loadings (0.75) among the five MetS components, followed by HDL-C (0.59), WC (0.47), MAP (0.30), and FBG (0.25) (Table 2). There were significant differences in the factor loadings of the MetS components across the eight subgroups, except for FBG ($P = 0.115$). TG had higher factor loadings in the Hans (ranging from 0.74–0.84) than minorities (0.58–0.78), especially with sex and age being equal. Similarly, HDL-C had higher factor loadings in the Hans (0.60–0.74) than minorities (0.41–0.65), also in the old minorities (0.62–0.65). WC had generally higher factor loadings in males (0.46–0.56) than females (0.34–0.40), especially in male minorities (0.55–0.56). Contrary to TG and HDL-C, MAP had higher factor loadings in the minorities (0.20–0.31) than Hans (0.11–0.27), and such disparities were more apparent in old people (0.27 vs 0.15 in males, and 0.20 vs. 0.11 in females). FBG had higher factor loadings in the young (0.20–0.27) than old (0.13–0.22), especially in the minorities (0.20 vs. 0.13 in males, and 0.27 vs. 0.13 in females). Model 2 with varying factor loadings of the MetS components across the demographic subgroups outperformed Model 1, resulting in a smaller overall AIC and some age-sex-ethnicity-specific submodels with a better fit to the data, such as for Hans (especially old Hans) and old minorities.

The equations fitted for the overall population and age-sex-ethnicity-specific subpopulations were presented (Table 3), for calculating a given individual's MetS score. An online tool was developed and made available to the public, allowing the easy clinical use by entering actual values of the MetS components to the equations (https://gisersqdai.shinyapps.io/mets_severiity_calculator/).

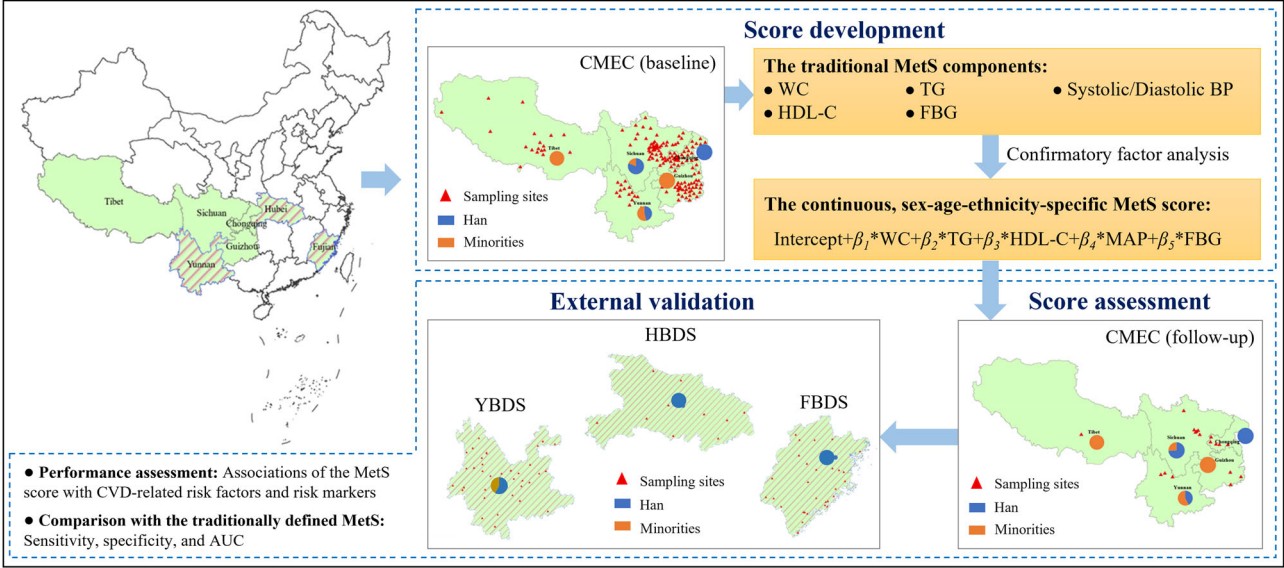

**Fig. 1 | An overview of the study design.** AUC area under the receiver operating characteristic (ROC) curve, BP blood pressure, CMEC China Multi-Ethnic Cohort, CVD cardiovascular disease, FBDS Fujian Behavior and Disease Surveillance cohort, FBG fasting blood glucose, HDL-C high-density lipoprotein cholesterol, HBDS Hubei Behavior and Disease Surveillance cohort, MAP mean arterial pressure, MetS metabolic syndrome, TG triglycerides, WC waist circumference, YBDS Yunnan Behavior and Disease Surveillance cohort. Reprinted from the National Geographic Information Resources Directory Service System https://www.webmap.cn/commres.do?method=result100W.

**Table 1 | The prevalence of metabolic syndrome (MetS) and levels of the MetS components among the China Multi-Ethnic Cohort (CMEC) participants, by sex, age, and ethnicity**

| | n | N of MetS (%) | Mean ± SD | | | | |
| --- | --- | --- | --- | --- | --- | --- | --- |
| | | | WC (cm) | TG (mmol/L) | HDL-C (mmol/L) | MAP (mmHg) | FBG (mmol/L) |
| Overall | 77,639 | 15,026 (19.4) | 81.8 ± 10.1 | 1.6 ± 1.5 | 1.5 ± 0.4 | 93.7 ± 12.5 | 5.2 ± 1.1 |
| Han | | | | | | | |
| Male | | | | | | | |
| <60 | 14,651 | 3,510 (24.0) | 84.5 ± 9.3 | 2.0 ± 1.9 | 1.3 ± 0.4 | 96.1 ± 11.9 | 5.3 ± 1.3 |
| ≥60 | 4,712 | 949 (20.1) | 82.7 ± 9.3 | 1.4 ± 1.1 | 1.5 ± 0.4 | 98.7 ± 12.5 | 5.5 ± 1.1 |
| Female | | | | | | | |
| <60 | 20,611 | 2,637 (12.8) | 77.6 ± 8.7 | 1.4 ± 1.2 | 1.5 ± 0.4 | 89.7 ± 11.5 | 5.1 ± 0.8 |
| ≥60 | 4,808 | 1,450 (30.2) | 81.5 ± 9.6 | 1.7 ± 1.1 | 1.6 ± 0.4 | 96.5 ± 12.2 | 5.5 ± 1.0 |
| Minority | | | | | | | |
| Male | | | | | | | |
| <60 | 8,186 | 1,973 (24.1) | 86.7 ± 10.5 | 2.0 ± 2.1 | 1.4 ± 0.4 | 96.5 ± 12.5 | 5.2 ± 1.4 |
| ≥60 | 2,746 | 528 (19.2) | 83.3 ± 10.6 | 1.5 ± 1.2 | 1.5 ± 0.4 | 99.7 ± 13.3 | 5.4 ± 1.2 |
| Female | | | | | | | |
| <60 | 17,833 | 2,949 (16.5) | 81.8 ± 10.4 | 1.4 ± 1.2 | 1.5 ± 0.4 | 91.3 ± 11.9 | 5.1 ± 0.9 |
| ≥60 | 4,092 | 1,032 (25.2) | 82.3 ± 11.0 | 1.6 ± 1.0 | 1.6 ± 0.4 | 96.9 ± 12.9 | 5.3 ± 1.1 |
| p value[a] | | | 0.013 | <0.001 | <0.001 | <0.001 | <0.001 |

FBG fasting blood glucose, HDL-C high-density lipoprotein cholesterol, MAP mean arterial pressure, SD standard deviation, TG triglycerides, WC waist circumference.
[a]Significance of the differences in the values among the eight age-sex-ethnicity-specific subgroups, tested by the two-sided analysis of variance.

**Table 2 | Factor loadings of the metabolic syndrome (MetS) components and model fit assessments**

| MetS components | Model 1 | Model 2 | | | | | | | | | |
| --- | --- | --- | --- | --- | --- | --- | --- | --- | --- | --- | --- |
| | Overall | Overall | Han | | | | Minority | | | | p value[a] |
| | | | Male | | Female | | Male | | Female | | |
| | | | <60 | ≥60 | <60 | ≥60 | <60 | ≥60 | <60 | ≥60 | |
| WC | 0.47 | - | 0.48 | 0.46 | 0.38 | 0.35 | 0.55 | 0.56 | 0.34 | 0.40 | 0.004 |
| TG | 0.75 | - | 0.81 | 0.76 | 0.84 | 0.74 | 0.61 | 0.58 | 0.78 | 0.63 | <0.001 |
| HDL-C | 0.59 | - | 0.62 | 0.66 | 0.60 | 0.74 | 0.48 | 0.62 | 0.41 | 0.65 | <0.001 |
| MAP | 0.30 | - | 0.27 | 0.15 | 0.27 | 0.11 | 0.31 | 0.27 | 0.28 | 0.20 | 0.006 |
| FBG | 0.25 | - | 0.26 | 0.22 | 0.24 | 0.17 | 0.20 | 0.13 | 0.27 | 0.13 | 0.115 |
| **Model fit indexes** | | | | | | | | | | | |
| Chi-square/df | 1601.663 | 207.095 | | | | | | | | | |
| AIC | | | | | | | | | | | |
| 1,053,873.962 | | | | | | | | | | | |
| **1,024,731.526** | | | | | | | | | | | |
| RMSEA | 0.144 | 0.146 | **0.142** | **0.130** | **0.139** | **0.117** | 0.176 | **0.134** | 0.158 | **0.125** | |
| SRMR | 0.064 | 0.060 | 0.066 | 0.064 | 0.069 | **0.062** | 0.083 | **0.060** | 0.077 | **0.058** | |
| CFI | 0.824 | 0.804 | **0.853** | **0.855** | **0.841** | **0.873** | 0.676 | **0.805** | 0.699 | 0.803 | |
| GFI | 0.962 | 0.962 | 0.961 | **0.966** | 0.962 | **0.972** | 0.942 | **0.966** | 0.954 | **0.970** | |
| MFI | 0.950 | 0.948 | **0.951** | **0.959** | **0.953** | **0.966** | 0.926 | **0.956** | 0.939 | **0.962** | |
| NFI | 0.824 | 0.803 | **0.852** | **0.854** | **0.841** | **0.872** | 0.675 | 0.803 | 0.699 | 0.801 | |

AIC Akaike's Information Criteria, CFI Comparative Fit Index, GFI Goodness of Fit Index, FBG fasting blood glucose, HDL-C high-density lipoprotein cholesterol, MAP mean arterial pressure, MFI Mc Donald's Fit Index, NFI Bentler-Bonett Normed Fit Index, RMSEA Root Mean Square Error of Approximation, SRMR Standardized Root Mean Square Residual, TG triglycerides, WC waist circumference.
The values of indexes of model fit in Model 2 were in bold, if better than the ones in Model 1.
[a]Significance of the differences in factor loadings among the eight age-sex-ethnicity-specific subgroups, estimated by the two-sided Chi-square test.

## Performance assessment of the MetS score

In the CMEC baseline, each unit increase in the MetS score was associated with an increased risk for all CVD-related risk factors, including hyperlipidemia (OR = 1.30 [1.29, 1.32]), diabetes (OR = 1.05 [1.05, 1.05]) and hypertension (OR = 1.09 [1.07, 1.11]), and also an elevated level of all CVD-related risk markers, including HbA1c ($\beta$ = 0.17 [0.10, 0.17]), CHOL ($\beta$ = 0.24 [0.23, 0.24]), BMI ($\beta$ = 1.71 [1.69, 1.73]), and SUA

($\beta$ = 26.21 [25.70, 26.72]) (Table S2). There were heterogeneities in these associations among the age-sex-ethnicity-specific subgroups. For example, the increase of the risk for hyperlipidemia per unit increase in the MetS score was smallest among old female minorities (OR = 1.20 [1.12, 1.29]) and largest among young male Hans (OR = 1.34 [1.31, 1.38]); similarly, the increase of the CHOL level per unit increase in the MetS score was smallest among old female minorities ($\beta$ = 0.05

**Table 3 | Equations of the age-sex-ethnicity-specific metabolic syndrome (MetS) scores**

| Groups | Equations |
|---|---|
| Overall | −3.1436 + 0.0258*WC + 0.361*TG-0.9348*HDL-C + 0.0128*MAP + 0.1224*FBG |
| Han | |
| Male | |
| <60 | −2.9092 + 0.0262*WC + 0.3098*TG-0.944*HDL-C + 0.0097*MAP + 0.0745*FBG |
| ≥60 | −2.3741 + 0.0264*WC + 0.4933*TG-0.999*HDL-C + 0.0054*MAP + 0.0821*FBG |
| Female | |
| <60 | −2.4981 + 0.0199*WC + 0.5218*TG-0.8616*HDL-C + 0.0110*MAP + 0.1074*FBG |
| ≥60 | −0.5682 + 0.0153*WC + 0.4587*TG-1.3567*HDL-C + 0.0036*MAP + 0.0688*FBG |
| Minority | |
| Male | |
| <60 | −5.0941 + 0.0427*WC + 0.2038*TG-0.9073*HDL-C + 0.0172*MAP + 0.1101*FBG |
| ≥60 | −3.7784 + 0.0399*WC + 0.3234*TG-1.0218*HDL-C + 0.0116*MAP + 0.0738*FBG |
| Female | |
| <60 | −3.6762 + 0.0189*WC + 0.5237*TG-0.6562*HDL-C + 0.0147*MAP + 0.2064*FBG |
| ≥60 | −1.5436 + 0.0212*WC + 0.4698*TG-1.3471*HDL-C + 0.0084*MAP + 0.0821*FBG |

*FBG* fasting blood glucose (mmol/L), *HDL-C* high-density lipoprotein cholesterol (mmol/L), *MAP* mean arterial pressure (mmHg), *TG* triglycerides (mmol/L), *WC* waist circumference (cm).

**Table 4 | Associations between the metabolic syndrome (MetS) score and the cardiovascular disease (CVD)-related risk factors and risk markers in the China Multi-Ethnic Cohort (CMEC) follow-up survey**

| Groups | HR (95% CI) | | | β (95% CI) | | | |
|---|---|---|---|---|---|---|---|
| | Hyperlipidemia | Diabetes | Hypertension | HbA1c | CHOL | BMI | SUA |
| Overall[a] | 1.72 (1.69, 1.75)*** | 1.61 (1.45, 1.78)*** | 1.42 (1.39, 1.45)*** | 0.16 (0.14, 0.18)*** | 0.18 (0.17, 0.20)*** | 1.62 (1.56, 1.68)*** | 24.94 (23.73, 26.16)*** |
| Han[b] | | | | | | | |
| Male | | | | | | | |
| <60 | 1.74 (1.68, 1.79)*** | 1.60 (1.39, 1.85)*** | 1.38 (1.32, 1.44)*** | 0.14 (0.10, 0.17)*** | 0.18 (0.14, 0.22)*** | 1.54 (1.39, 1.69)*** | 27.32 (24.24, 30.41)*** |
| ≥60 | 2.06 (1.91, 2.22)*** | 1.49 (1.30, 1.70)*** | 1.16 (1.08, 1.24)*** | 0.14 (0.09, 0.19)*** | 0.20 (0.12, 0.28)*** | 1.34 (1.18, 1.50)*** | 17.51 (13.47, 21.56)*** |
| Female | | | | | | | |
| <60 | 2.06 (1.98, 2.15)*** | 1.86 (1.69, 2.05)*** | 1.51 (1.44, 1.59)*** | 0.17 (0.14, 0.20)*** | 0.18 (0.15, 0.22)*** | 1.34 (1.24, 1.44)*** | 20.59 (18.65, 22.54)*** |
| ≥60 | 1.72 (1.56, 1.88)*** | 1.41 (1.19, 1.65)*** | 1.12 (1.03, 1.22)** | 0.10 (0.05, 0.15)*** | −0.01 (−0.10, 0.07) | 1.10 (0.90, 1.30)*** | 17.30 (13.01, 21.60)*** |
| Minority[b] | | | | | | | |
| Male | | | | | | | |
| <60 | 2.05 (1.91, 2.21)*** | 1.88 (1.53, 2.31)*** | 1.79 (1.63, 1.96)*** | 0.19 (0.13, 0.25)*** | 0.23 (0.18, 0.28)*** | 2.30 (2.16, 2.43)*** | 31.66 (27.76, 35.56)*** |
| ≥60 | 2.13 (1.80, 2.53)*** | 2.04 (1.40, 2.96)*** | 1.40 (1.21, 1.61)*** | 0.20 (0.08, 0.32)*** | 0.11 (0.01, 0.20)* | 2.31 (2.06, 2.55)*** | 24.04 (15.61, 32.47)*** |
| Female | | | | | | | |
| <60 | 2.02 (1.93, 2.13)*** | 1.88 (1.66, 2.13)*** | 1.56 (1.46, 1.67)*** | 0.11 (0.09, 0.14)*** | 0.23 (0.19, 0.26)*** | 1.17 (1.06, 1.28)*** | 19.01 (16.86, 21.16)*** |
| ≥60 | 1.54 (1.41, 1.68)*** | 1.53 (1.30, 1.80)*** | 1.26 (1.14, 1.39)*** | 0.13 (0.06, 0.20)*** | 0.04 (−0.03, 0.12) | 1.37 (1.12, 1.62)*** | 25.81 (20.92, 30.71)*** |

*BMI* body mass index, *CHOL* cholesterol, *CI* confidence interval, *HbA1c* glycated hemoglobin, *HR* hazard ratios, *SUA* serum uric acid.
[a]Adjusted for age, sex, ethnicity, marital status, educational level, annual family income, residential location, alcohol drinking status, smoking status, diet, and physical activity at baseline.
[b]Adjusted for marital status, educational level, annual family income, residential location, alcohol drinking status, smoking status, diet, and physical activity at baseline.
HR and β are estimated by Cox regression and generalized estimating equations, respectively, and all tests are two-sided.
*p < 0.05, **p < 0.01, ***p < 0.001.

[0.01, 0.08]) and largest among old male minorities ($\beta = 0.30$ [0.29, 0.31]). Some other patterns included that the increase of the BMI level per unit increase in the MetS score was larger among the older in the Hans, but among the younger in the minorities.

In the CMEC follow-up (Table S3), each unit increase in the MetS score increased the rate of experiencing hyperlipidemia by 72% (HR = 1.72 [1.69, 1.75]), diabetes by 61% (HR = 1.61 [1.45, 1.78]), and hypertension by 42% (HR = 1.42), and also elevated the levels of HbA1c ($\beta = 0.16$ [0.14, 0.18]), CHOL ($\beta = 0.18$ [0.17, 0.20]), BMI ($\beta = 1.62$ [1.56, 1.68]), and SUA ($\beta = 24.94$ [23.73, 26.16]) (Table 4). Heterogeneities in

these associations were observed among the age-sex-ethnicity-specific subgroups. For example, among all age-sex-ethnicity-specific subgroups, the increases of the risk for hyperlipidemia (HR = 2.13 [1.80, 2.53]) and diabetes (HR = 2.04 [1.40, 2.96]) per unit increase in the MetS score were largest in old male minorities; the increases of the risk for diabetes (HR = 1.41 [1.19, 1.65]) and hypertension (HR = 1.12 [1.03, 1.22]) per unit increase in the MetS score were smallest in old female Hans. Some other patterns included that the increases of the risks for hyperlipidemia and diabetes and of the SUA level per unit increase in the MetS score were generally larger among the younger than their

older counterparts. The rates of having the CVD-related risk factors and risk markers increased as the quartiles of both the overall and age-sex-ethnicity-specific MetS scores, with the highest HRs observed in the 4th quartile of the MetS score for all groups (Table 5). The performance of the overall MetS score had moderate accuracy in predicting the CVD-related risk factors (C-index ranging from 0.71-0.80) and risk markers (C-index ranging from 0.75-0.76) except in predicting CHOL (C-index=0.65) (Table S4). The predictive performance of the MetS score in most age-sex-ethnicity-specific subgroups also had moderate accuracy, with the highest accuracy observed when predicting the elevated BMI among old male minorities (C-index=0.90).

### Comparison with the traditionally defined MetS
The overall and age-sex-ethnicity-specific MetS scores, after being converted into the dichotomous MetS variables, indicated the traditionally defined MetS with high accuracy except in the old Hans (both male and female) with highly acceptable accuracy (Fig. 2). The AUC value was 0.917 for the overall group, and ranged from 0.900 to 0.932 for the age-sex-ethnicity-specific subgroups other than the old male Hans (0.882) and old female Hans (0.869).

In comparison with the traditionally defined MetS, the dichotomous overall and age-sex-ethnicity-specific MetS scores showed higher sensitivity and AUC in predicting the presence of ≥1 CVD-related risk factor and ≥1 abnormal risk marker in the CMEC baseline (all $p$ values < 0.05), particularly in young male Hans (sensitivity = 0.61, AUC = 0.77), but showed the lower specificity (all $p$ values < 0.05) (Fig. 3, Table S5). The only exceptions were old female (both Hans and minorities) showing a comparable AUC between the traditionally defined MetS and the dichotomous MetS score.

### Performance assessment of the MetS score in three external dataset
The YBDS participants included about 69% aged <60, 45% being male, and 42% being minorities (Table S6). Each unit increase in the MetS score was associated with an increased risk for all CVD-related risk factors, including hyperlipidemia (OR = 1.31 [1.31, 1.32]), diabetes (OR = 1.08 [1.07, 1.08]), and hypertension (OR = 1.14 [1.13, 1.14]), and also an elevated level of all CVD-related risk markers, including HbA1c ($\beta = 0.22$ [0.21, 0.23]), CHOL ($\beta = 0.19$ [0.18, 0.20]), BMI ($\beta = 2.11$ [2.07, 2.15]), and SUA ($\beta = 24.57$ [23.65, 25.48]) (Table S7). Heterogeneities in these associations also existed among the age-sex-ethnicity-specific subgroups. For example, the increase of the risk for hyperlipidemia per unit increase in the MetS score was smallest among old female minorities (OR = 1.21 [1.19, 1.24]) and largest among young male Hans (OR = 1.38 [1.36, 1.40]), which were consistent with the patterns observed in the CMEC dataset. The risks of having the CVD-related risk factors and risk markers increased as the quartiles of both the overall and age-sex-ethnicity-specific MetS scores, with the highest ORs observed in the 4th quartile of the MetS score for all groups (Table S8).

The overall and age-sex-ethnicity-specific dichotomous MetS scores indicated the traditionally defined MetS with generally acceptable accuracy in the YBDS dataset. Some subgroups had highly acceptable accuracy, such as in young male Hans (0.865) and young female Hans (0.883) (Fig. 4). Compared to the traditionally defined MetS, the dichotomous overall and age-sex-ethnicity-specific MetS scores also showed the higher sensitivity and AUC in predicting the presence of ≥1 CVD-related risk factor and ≥1 abnormal risk marker (all $p$ values < 0.05) but showed the lower specificity (all $p$ values < 0.05) (Table S9). The only exceptions were old female Hans and old female minorities showing a comparable AUC between the traditionally defined MetS and the dichotomous MetS score.

The HBDS participants were all Hans, with about 66% aged <60 and 39% being male (Table S10). Each unit increase in the MetS score was associated with an increased risk for all CVD-related risk factors, including hyperlipidemia (OR = 1.35 [1.34, 1.35]), diabetes (OR = 1.13 [1.12, 1.14]), and hypertension (OR = 1.20 [1.19, 1.21]), and also an elevated level of all CVD-related risk markers, including CHOL ($\beta = 0.19$ [0.17, 0.20]), BMI ($\beta = 2.37$ [2.32, 2.41]), and SUA ($\beta = 29.34$ [27.85, 30.83]) (Table S11). The risks of having the CVD-related risk factors and risk markers increased as the quartiles of both the overall and age-sex-specific MetS scores, with the highest ORs observed in the 4th quartile of the MetS score for nearly all groups (Table S12). The traditionally defined MetS had generally acceptable accuracy in the HBDS dataset (Fig. 5), compared to which the dichotomous overall and age-sex-ethnicity-specific MetS scores showed the higher sensitivity and AUC in predicting the presence of ≥1 CVD-related risk factor and ≥1 abnormal risk marker, but showed the lower specificity (Fig. 5, Table S13).

The FBDS participants were all Hans as well, with about 66% aged <60 and 44% being male (Table S14). The similar performance was observed among the FBDS participants (Tables S15-S17, Fig. 5).

## Discussion
This study, for the first time, developed a continuous, age-sex-ethnicity-specific MetS score in the Chinese population based on the CMEC baseline survey. The MetS score had good performance in predicting the changes in CVD-related risk markers in the CMEC follow-up survey and the occurrence of CVD-related risk factors in both the CMEC follow-up and the three external surveys. This new MetS score offers improvements on the traditionally defined MetS to differentiate MetS severity, and had good performance in predicting the CVD risk among age-sex-ethnicity-specific subgroups in the Chinese population. Rather than replacing individual MetS components for decision-making in the choice of intervention and treatment strategies among those with abnormal MetS components, the MetS score would be mainly used for early identification of CVD risks among the general or healthy population. Particularly, if having the levels of all MetS components close to the corresponding cut-off values (on the normal side) but demonstrating no relevant symptoms, one would usually not get diagnosed and treated clinically but be at a greater risk for CVD. In such cases, those individuals could be identified due to their high MetS scores, and get further clinical examinations or lifestyle interventions to decrease the MetS score and thus CVD risks. Also, the MetS score may help determine the urgency of public health interventions in given populations, especially in those with higher risk and to be prioritized for timely clinical interventions. According to previous studies[7,8], people could be motivated by the MetS score and may feel empowered by a decrease in score.

Previous studies have developed MetS scores for the populations in developed, mostly western, countries. For example, adolescents' and adults' sex- and race/ethnicity-specific MetS scores were developed using the 1999–2010 data from 4413 adolescents and 6,870 adults in the National Health and Nutrition Examination Survey in the United States, where a significant correlation between the scores and biomarkers of future diseases was observed[7,14]. Another study in Singapore with 4419 participants developed an MetS score to assess the performance of predicting diabetes in the follow-up[15]. However, both scores were derived based on a single, small dataset without performance assessment in externally independent datasets, and both were in developed countries. The prevalence and components of MetS in the Chinese population are quite different from those of western people, mainly due to the differences in hereditary and lifestyle factors[10,16], which may result in the variation in the composition of MetS components and their weights in the models. The current age-sex-ethnicity specific MetS score was developed from a recently established large-sample, multi-ethnic cohort, and validated in the three newly established, independent large-sample datasets. Besides, the MetS score showed good predictive performance in detecting CVD-related risk factors and risk markers in the CMEC follow-up

**Table 5 | Associations between the metabolic syndrome (MetS) score in quartiles and the cardiovascular disease (CVD)-related risk factors and risk markers in the China Multi-Ethnic Cohort (CMEC) follow-up survey**

| Groups | Quartiles | HR (95% CI) | | | | | | |
|---|---|---|---|---|---|---|---|---|
| | | Hyperlipidemia | Diabetes | Hypertension | Elevated HbA1c | Elevated CHOL | Elevated BMI | Elevated SUA |
| Overall[a] | <-0.71 | 1.00 | 1.00 | 1.00 | 1.00 | 1.00 | 1.00 | 1.00 |
| | [-0.71, -0.11] | 1.92 (1.66, 2.23)*** | 1.64 (1.28, 2.11)*** | 1.80 (1.60, 2.04)*** | 1.77 (1.31, 2.38)*** | 1.62 (1.36, 1.93)*** | 1.19 (1.13, 1.25)*** | 2.10 (1.77, 2.48)*** |
| | [-0.11, 0.58] | 3.73 (3.26, 4.26)*** | 2.88 (2.29, 3.62)*** | 2.72 (2.42, 3.05)*** | 3.36 (2.55, 4.41)*** | 2.29 (1.94, 2.7)*** | 1.50 (1.42, 1.57)*** | 3.33 (2.85, 3.9)*** |
| | >0.58 | 11.26 (9.94, 12.75)*** | 6.64 (5.37, 8.22)*** | 4.01 (3.59, 4.48)*** | 7.18 (5.54, 9.31)*** | 2.90 (2.47, 3.41)*** | 1.91 (1.82, 2.01)*** | 5.63 (4.84, 6.55)*** |
| Han, male[b] | | | | | | | | |
| <60 | <-0.71 | 1.00 | 1.00 | 1.00 | 1.00 | 1.00 | 1.00 | 1.00 |
| | [-0.71, -0.08] | 3.13 (2.25, 4.33)*** | 1.97 (1.16, 3.32)* | 2.09 (1.61, 2.71)*** | 1.97 (1.06, 3.65)* | 1.80 (1.16, 2.80)** | 1.41 (1.23, 1.62)*** | 1.65 (1.27, 2.15)*** |
| | [-0.08, 0.60) | 7.24 (5.35, 9.81)*** | 3.57 (2.21, 5.77)*** | 3.24 (2.54, 4.14)*** | 3.68 (2.10, 6.44)*** | 2.85 (1.89, 4.31)*** | 1.91 (1.68, 2.17)*** | 2.20 (1.72, 2.82)*** |
| | ≥0.60 | 19.90 (14.85, 26.67)*** | 7.56 (4.85, 11.79)*** | 4.02 (3.17, 5.10)*** | 7.62 (4.52, 12.87)*** | 3.30 (2.20, 4.96)*** | 2.37 (2.10, 2.68)*** | 4.00 (3.17, 5.05)*** |
| ≥60 | <-0.70 | 1.00 | 1.00 | 1.00 | 1.00 | 1.00 | 1.00 | 1.00 |
| | [-0.70, -0.12) | 1.91 (1.04, 3.50)* | 1.51 (0.85, 2.69) | 1.42 (1.10, 1.84)** | 1.50 (0.77, 2.91) | 1.61 (0.79, 3.29) | 1.16 (0.92, 1.45) | 1.56 (0.96, 2.53) |
| | [-0.12, 0.58) | 4.09 (2.41, 6.95)*** | 2.23 (1.30, 3.84)** | 1.78 (1.39, 2.28)*** | 2.18 (1.17, 4.07)* | 3.83 (2.12, 6.90)*** | 1.45 (1.15, 1.82)* | 1.75 (1.08, 2.85)* |
| | ≥0.58 | 14.64 (9.04, 23.71)*** | 3.90 (2.40, 6.35)*** | 1.78 (1.40, 2.26)*** | 3.56 (2.01, 6.3)*** | 4.03 (2.20, 7.41)*** | 2.08 (1.71, 2.54)*** | 3.00 (1.95, 4.61)*** |
| Han, female[b] | | | | | | | | |
| <60 | <-0.70 | 1.00 | 1.00 | 1.00 | 1.00 | 1.00 | 1.00 | 1.00 |
| | [-0.70, -0.12) | 2.74 (1.94, 3.88)*** | 2.57 (1.52, 4.34)*** | 2.01 (1.49, 2.71)*** | 3.64 (1.60, 8.28)** | 2.75 (1.85, 4.11)*** | 1.11 (1.01, 1.23)** | 1.40 (0.98, 1.99) |
| | [-0.12, 0.55) | 4.31 (3.10, 5.98)*** | 3.27 (1.97, 5.43)*** | 3.22 (2.43, 4.26)*** | 6.06 (2.81, 13.09)*** | 3.73 (2.52, 5.50)*** | 1.35 (1.24, 1.48)*** | 2.48 (1.79, 3.43)*** |
| | ≥0.55 | 14.97 (11.04, 20.32)*** | 8.50 (5.36, 13.46)*** | 5.05 (3.86, 6.61)*** | 15.84 (7.66, 32.73)*** | 4.88 (3.35, 7.11)*** | 1.62 (1.48, 1.77)*** | 4.57 (3.36, 6.21)*** |
| ≥60 | <-0.72 | 1.00 | 1.00 | 1.00 | 1.00 | 1.00 | 1.00 | 1.00 |
| | [-0.72, -0.09) | 0.99 (0.66, 1.48) | 1.07 (0.58, 2.00) | 0.90 (0.68, 1.19) | 1.11 (0.55, 2.22) | 0.83 (0.54, 1.27) | 1.13 (0.94, 1.36) | 0.88 (0.49, 1.58) |
| | [-0.09, 0.58) | 1.46 (0.99, 2.14) | 1.94 (1.10, 3.41)* | 1.22 (0.94, 1.59) | 2.13 (1.14, 4.00)* | 1.09 (0.72, 1.67) | 1.35 (1.13, 1.61)*** | 1.72 (1.01, 2.92)* |
| | ≥0.58 | 3.74 (2.65, 5.27)*** | 2.32 (1.32, 4.07)** | 1.29 (1.01, 1.66)* | 2.15 (1.14, 4.04)* | 1.06 (0.69, 1.64) | 1.41 (1.19, 1.68)*** | 2.64 (1.59, 4.38)*** |
| Minority, male[b] | | | | | | | | |
| <60 | <-0.72 | 1.00 | 1.00 | 1.00 | 1.00 | 1.00 | 1.00 | 1.00 |
| | [-0.72, -0.09) | 1.59 (1.09, 2.31)* | 1.52 (0.61, 3.81) | 2.69 (1.70, 4.24)*** | 1.75 (0.67, 4.59) | 1.72 (0.98, 3.01) | 1.43 (1.15, 1.78)* | 3.00 (1.97, 4.57)*** |
| | [-0.09, 0.58) | 3.61 (2.64, 4.92)*** | 2.08 (0.91, 4.81) | 3.28 (2.12, 5.07)*** | 2.07 (0.85, 5.07) | 1.76 (1.03, 3.00)* | 2.28 (1.89, 2.75)*** | 4.21 (2.82, 6.30)*** |
| | ≥0.58 | 7.38 (5.53, 9.84)*** | 6.35 (3.03, 13.28)*** | 6.31 (4.20, 9.46)*** | 6.23 (2.77, 13.99)*** | 3.54 (2.21, 5.66)*** | 3.03 (2.54, 3.60)*** | 6.13 (4.15, 9.06)*** |
| ≥60 | <-0.73 | 1.00 | 1.00 | 1.00 | 1.00 | 1.00 | 1.00 | 1.00 |
| | [-0.73, -0.04) | 2.21 (0.93, 5.24) | 1.69 (0.59, 4.86) | 1.36 (0.84, 2.20) | 1.94 (0.62, 6.13) | 2.81 (0.91, 8.66) | 1.43 (1.00, 2.06) | 1.22 (0.61, 2.44) |
| | [-0.04, 0.66) | 3.55 (1.60, 7.87)** | 1.91 (0.69, 5.33) | 1.78 (1.13, 2.82)* | 2.13 (0.68, 6.66) | 3.55 (1.29, 9.74)* | 1.92 (1.35, 2.74)*** | 1.78 (0.91, 3.46) |
| | ≥0.66 | 9.20 (4.35, 19.46)*** | 4.87 (1.87, 12.69)** | 2.20 (1.40, 3.44)*** | 4.66 (1.63, 13.36)** | 4.20 (1.42, 12.39)** | 2.85 (2.02, 4.02)*** | 3.07 (1.65, 5.73)*** |

**Table 5 (continued) | Associations between the metabolic syndrome (MetS) score in quartiles and the cardiovascular disease (CVD)-related risk factors and risk markers in the China Multi-Ethnic Cohort (CMEC) follow-up survey**

| Groups | Quartiles | HR (95% CI) | | | | | | | | |
|---|---|---|---|---|---|---|---|---|---|---|
| | | Hyperlipidemia | Diabetes | Hypertension | Elevated HbA1c | Elevated CHOL | Elevated BMI | Elevated SUA |
| Minority, female[b] | | | | | | | | |
| <60 | <-0.70 | 1.00 | 1.00 | 1.00 | 1.00 | 1.00 | 1.00 | 1.00 |
| | [-0.70, -0.10) | 2.12 (1.54, 2.92)*** | 2.16 (0.93, 5.02) | 1.57 (1.10, 2.24)* | 2.22 (0.91, 5.37) | 1.68 (1.09, 2.58)* | 1.09 (0.99, 1.21) | 2.17 (1.42, 3.31)*** |
| | [-0.10, 0.59) | 3.06 (2.25, 4.16)*** | 3.78 (1.73, 8.24)*** | 3.08 (2.22, 4.27)*** | 3.54 (1.54, 8.12)** | 2.24 (1.47, 3.41)*** | 1.21 (1.09, 1.34)*** | 3.27 (2.18, 4.90)*** |
| | ≥0.59 | 9.44 (7.17, 12.45)*** | 9.84 (4.80, 20.16)*** | 4.24 (3.08, 5.84)*** | 9.55 (4.46, 20.44)*** | 3.99 (2.71, 5.87)*** | 1.38 (1.25, 1.51)*** | 6.67 (4.61, 9.64)*** |
| ≥60 | <-0.72 | 1.00 | 1.00 | 1.00 | 1.00 | 1.00 | 1.00 | 1.00 |
| | [-0.72, -0.09) | 0.51 (0.30, 0.86)* | 0.97 (0.34, 2.78) | 1.72 (1.08, 2.75)* | 1.00 (0.33, 3.05) | 0.42 (0.23, 0.78)** | 1.39 (1.10, 1.74)* | 3.70 (1.12, 12.17)* |
| | [-0.09, 0.59) | 1.26 (0.83, 1.90) | 3.39 (1.36, 8.45)** | 2.32 (1.46, 3.69)*** | 3.73 (1.41, 9.84)** | 1.00 (0.63, 1.61) | 1.58 (1.27, 1.97)*** | 8.77 (3.10, 24.83)*** |
| | ≥0.59 | 2.57 (1.82, 3.63)*** | 5.07 (2.15, 11.93)*** | 2.65 (1.70, 4.12)*** | 4.73 (1.92, 11.63)*** | 0.99 (0.64, 1.52) | 1.66 (1.33, 2.08)*** | 9.30 (3.31, 26.17)*** |

BMI body mass index, CHOL cholesterol, CI confidence interval, HbA1c glycated hemoglobin, HR hazard ratios, SUA serum uric acid. [a]Adjusted for age, sex, ethnicity, marital status, educational level, annual family income, residential location, alcohol drinking, smoking, diet, and physical activity at baseline. [b]Adjusted for marital status, educational level, annual family income, residential location, alcohol drinking, smoking, diet, and physical activity at baseline. HR is estimated by Cox regression, and all tests are two-sided. *p < 0.05, **p < 0.01, ***p < 0.001.

survey, as well as in the external datasets, which has demonstrated the generalizability of the MetS score.

The MetS score was developed mainly based on the traditional MetS components, resulting in the higher weights of TG, HDL-C, and WC. TG and HDL-C have been considered surrogate insulin resistance in the diagnosis of MetS[17,18], and their relationships with insulin resistance differ by ethnicity;[18] WC may to a large extent indicate the deposition of excess fat in internal organs, which could increase MetS and CVD risks by disrupting hormone secretion, lowing the level of insulin, and elevate the level of blood fat[19]. One prospective study in China has also showed that HDL-C, TG, and WC were better predictors of MetS than BP and FBG[20], which supported the patterns of weights of MetS components found in this study. MAP and FBG exhibited the lower weights than other components, which was not surprising given the relatively high rates of essential hypertension and diabetes independent of MetS[21]. Previous studies have also revealed the low loading factors (<0.3) of SBP and FBG[7,14,15]. However, they remained in the equations because they can provide an additive effect on the capacity of the MetS score for assessing CVD risks.

Notably, we observed that the factor loadings of the MetS components varied greatly by age, sex, and ethnicity, suggesting that the MetS scores for different subgroups should be considered, such as the high weights of TG and HDL-C in the Hans, relatively higher weight of WC in minorities and males, and higher weight of TG in young adults. The variations in the age-sex-ethnicity-specific MetS score may also be explained by central obesity and lipid measurement across different populations. The high level of blood lipid markers may be more closely associated with MetS in Hans than minorities[10], which may account for the greater weights of TG and HDL in the Hans. The weight of TG in young people may partly be explained by the aging effect on the complex pathophysiological mechanisms of metabolic diseases, and blood lipids are considered precursors to insulin resistance and closer to MetS among young adults[22–24]. Moreover, the higher weight of WC in males suggested that central obesity may be a more sensitive marker of MetS in males. Unlike in other countries, central obesity in China has been increasingly prevalent in males in recent years. The higher WC in minorities may be related to their lifestyles and ambient natural environment, compared to the Hans[25]. For example, central obesity in minorities may be attributed to low temperature, high altitude, high-fat food, and exercise-discouraging environments[26–29].

Compared to the traditionally defined MetS, the MetS score, both in the continuous form or in quartiles, had the higher sensitivity and AUC in diagnosing CVD-related risk factors and risk markers. This has strengthened its use in the management of CVD-related risk factors associated with the population-level and individual-level characteristics. Specifically, at the individual level, with the MetS score calculator as a freely available online tool, healthcare providers can estimate an individual's chance of developing CVD-related risk factors in the future. Changes in the MetS score over time would suggest an individual's worsening risk, which may trigger the motivation of intervention or therapies at early stages. At the population level, the MetS score can identify the future CVD risks in a given population, identify the high-risk groups, and determine the urgency and priority of public health interventions. Previous studies have assessed the changes in the American sex-ethnicity-specific MetS score over time, and reported that an increased score may draw attention to the altered disease risk and help tracking earlier response to treatment closely[8,30]. As the burden of CVD in China increases and spreads, the web-based MetS score calculator could assist medical workers in applying prevention practices among those with a high MetS score.

Several limitations in this study should be noted. First, some age-sex-ethnicity-specific subgroups have a relatively small sample size, such as old male minorities, resulting in some less-than-optimal performance. However, to the knowledge of the authors, the sample size of the most subgroups based upon which the MetS score has been

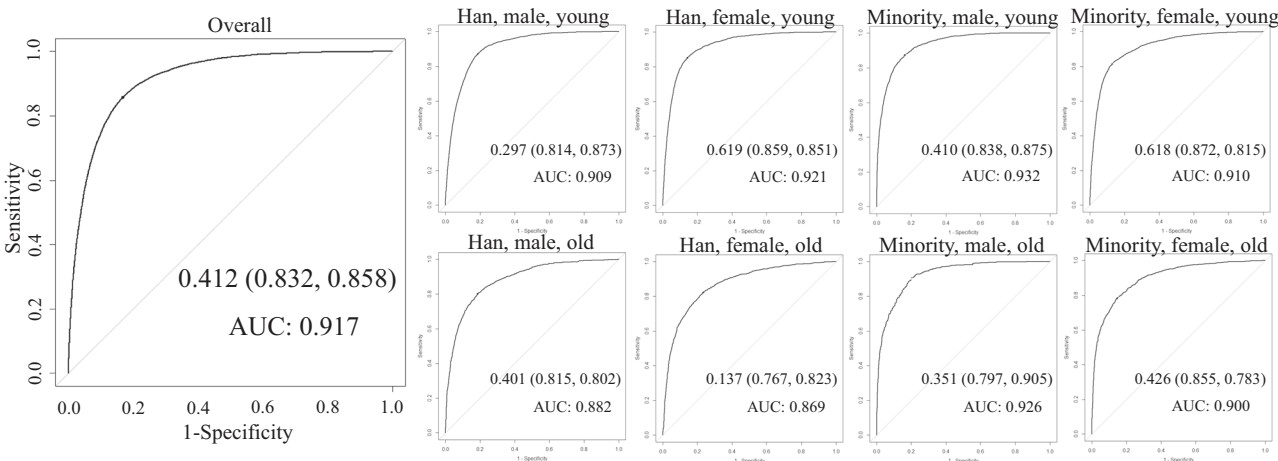

**Fig. 2 | Receiver operating characteristic (ROC) curves for the overall and age-sex-ethnicity-specific metabolic syndrome (MetS) scores and the traditionally defined MetS in the China Multi-Ethnic Cohort (CMEC) dataset.** AUC area under the ROC curve.

**Fig. 3 | Comparison of the traditionally defined metabolic syndrome (MetS) and the dichotomous MetS score in detecting the cardiovascular disease (CVD)-related risk factors and risk markers in the CMEC dataset.** The CVD-related risk factors included hypertension, diabetes, and dyslipidaemia; the CVD-related risk markers included elevated glycated hemoglobin (HbA1c), cholesterol, body mass index, and serum uric acid. *Significance ($p < 0.05$) for the differences in the capacity of detecting the CVD-related risk factors and risk markers by the traditionally defined MetS and the dichotomous MetS score, with the paired Chi-square test used for sensitivity and specificity and the Delong's test used for area under the curve (AUC). All tests are two-sided. The error bars indicate 95% confidence intervals of values of sensitivity, specificity, and AUC (see Table S5 for all specific values).

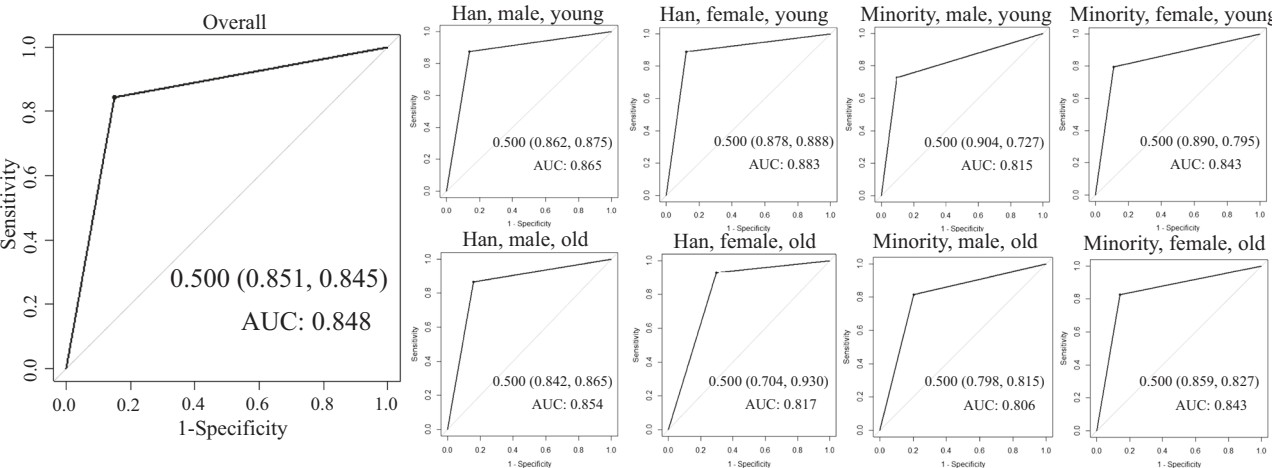

**Fig. 4 | Receiver operating characteristic (ROC) curves for the dichotomous overall and age-sex-ethnicity-specific MetS scores and the traditionally defined MetS in the Yunnan Behavior and Disease Surveillance (YBDS) survey.** AUC area under the ROC curve.

developed and validated was quite large, particularly much larger than that of the existing study populations elsewhere. Also, Hans in this study, accounting for 57.7% of the study population, were disproportional to the Hans in the study area. Nonetheless, the MetS score of the Hans has been validated externally (all Hans in the HBDS and FBDS), which implies that the large sample size may cancel out potential under-representation. Future efforts to further validate the MetS score in larger, specific populations are warranted. Second, some risk markers were not included for performance assessment due to the lack of measurement, such as high-sensitivity C-reactive protein (hs-CRP) and homeostasis model assessment of insulin resistance (HOMA-IR), so this score should not be used without caution. Third, although lifestyles and dietary patterns were adjusted in the models, the study participants were limited to western China, where lifestyles and dietary patterns may be different from other places in China. Such differences may still slightly affect our results. Also, the participants without information on MetS were excluded without being able to conduct imputation. However, the MetS score was developed on the basis of a large sample and validated externally, it is reasonable to assume that missing data did not affect our results to a large extent. Fourth, although outside the scope of this study, heterogeneities across all 55 minority ethnic groups in China were not examined due to insufficient sample size in most groups, which is expected to be studied in future efforts. Such inter-minority differences should not be ignored whenever possible, and considering these differences would further improve the quality of the MetS score.

In conclusion, this study provides a continuous, robust, age-sex-ethnicity-specific MetS score to reflect the severity of MetS and predict CVD risks in different demographic subpopulations. An online MetS score calculator was developed and made available, which holds potential to help primary medical workers quantify CVD risk quickly and conveniently, and inform choice of medication and non-pharmacological interventions for chronic disease management. This continuous MetS score would also be beneficial to monitor the development of MetS and to improve overall population health in China and in other eastern countries. More broadly, this study would advance preventive medicine and sustainable development goals (SDGs).

## Methods
The CMEC was approved by the medical ethics committee of Sichuan University (K2016038). The three provincial surveys received ethical approval from the ethical review board of the Yunnan Center for Disease Prevention and Control (202017), the Fujian Center for Disease

Prevention and Control (2018001), and the National Center for Chronic and Non-communicable Disease Control and Prevention. All participants provided written informed consent before the survey.

### Study population and design
The CMEC study is an ongoing community-based prospective cohort study, which aims to examine ethnic variation in the profiles of non-communicable diseases and related risk factors in China[13]. The CMEC used a multi-stage stratified cluster sampling method to recruit 99,556 participants aged 30–79 years from the five provinces of Southwest China (Sichuan, Chongqing, Yunnan, Guizhou, and Xizang) between May 2018 and September 2019. Among them, about 10% of the subjects in each participating district/county, adding up to 11,527 participants, were selected between August 2020 and July 2021 by a purposive sampling method and followed up in the same way as the baseline survey.

Three provincially representative epidemiological surveys established in Yunnan Province (the one with the largest number of minorities in China), Hubei Province, and Fujian Province, were used to validate the MetS score. Among them, the Yunnan Behavior and Disease Surveillance cohort (YBDS) recruited 51,480 adults aged >18 years from 35 districts/counties of Yunnan Province from January to August 2021, by a multi-stage stratified cluster random sampling method; similarly, during 2018-2020, the Hubei Behavior and Disease Surveillance cohort (HBDS) recruited 27,964 adults aged >18 years from 10 districts/counties of Hubei Province, and the Fujian Behavior and Disease Surveillance cohort (FBDS) recruited 54,961 adults aged >18 years from 29 districts/counties of Fujian Province.

An overview of the study design is shown in Fig. 1. The baseline data of the eligible CMEC participants were used to develop the MetS score. All with complete information on the MetS components and sociodemographic characteristics (e.g., sex, age, and ethnicity) were included. Those with self-reported hypertension, diabetes, or hyper-lipidaemia and taking relevant medication at the time of the survey were excluded, as medication taken by them to control blood pressure, blood glucose, or lipids may make their measurements not capable of reflecting the natural levels. A total of 77,639 participants were included, with 21,917 (22.0%) excluded who had missed any MetS component and were taking antihyperlipidemic, anti-diabetic, or antihypertensive medications (Table S1). The CMEC follow-up data were used for internal validation of the MetS score, where the 9,249 participants were included after excluding those with incomplete information on the MetS components and diagnosed as CVD-related risk factors at baseline (Table S3). The YBDS, HBDS, and FBDS data

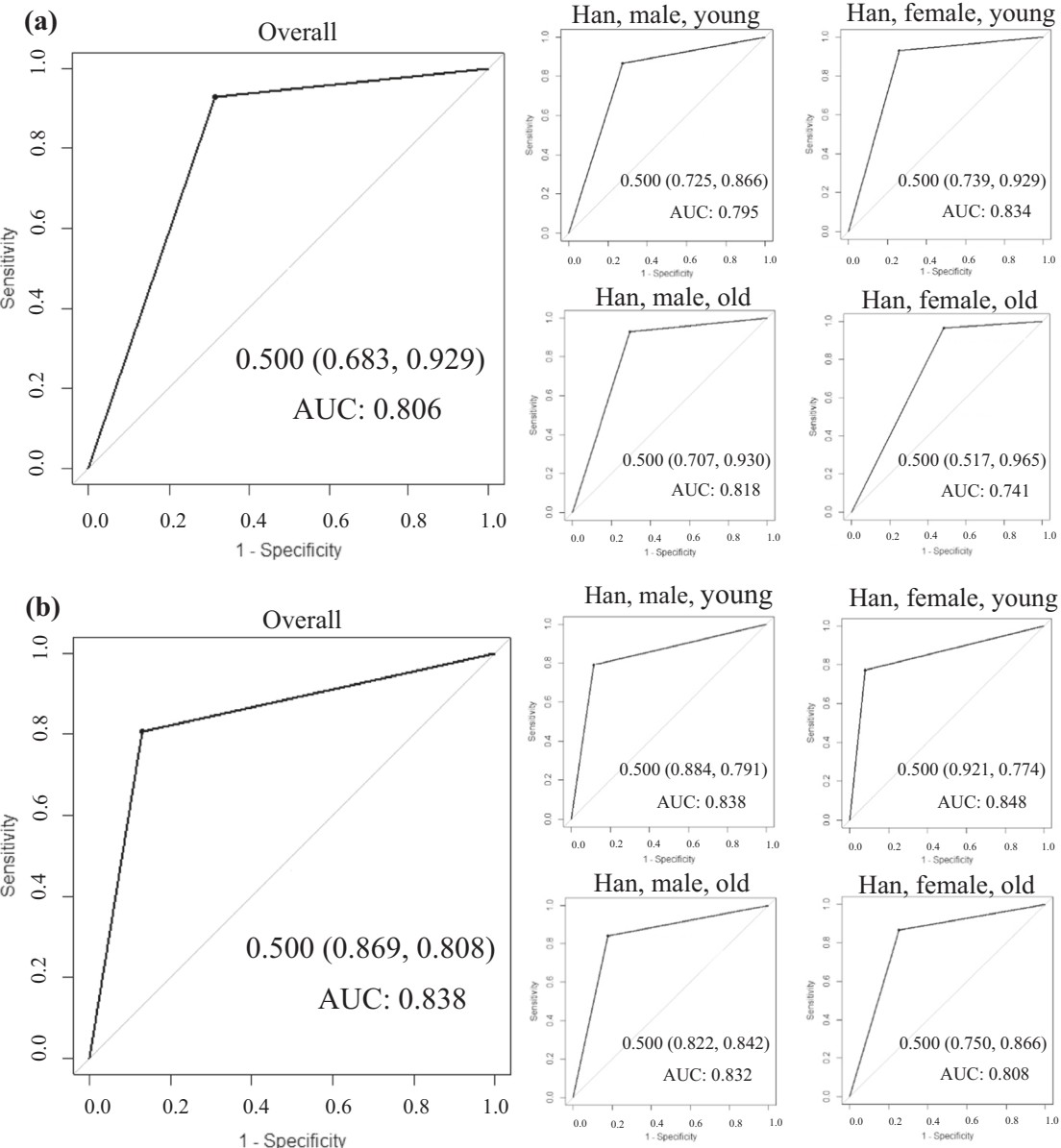

**Fig. 5 | Receiver operating characteristic (ROC) curves for the dichotomous overall and age-sex-ethnicity-specific MetS scores and the traditionally defined MetS in the (a) Hubei Behavior and Disease Surveillance (HBDS) survey and (b)** Fujian Behavior and Disease Surveillance (FBDS) survey. AUC area under the ROC curve.

were used for external validation, where a total of 99,563 participants were included after excluding those with incomplete information on the MetS components, aged <30 years, or taking antihyperlipidemic, anti-diabetic, or antihypertensive medications (Tables S6, S10, S14). A flowchart of participant inclusion and exclusion can be found in Figure S1.

## Data collection

The surveys of the CMEC, YBDS, HBDS, and FBDS consisted of an electronic questionnaire with face-to-face interviews, medical physical examinations, and clinical laboratory tests. The information on sociodemographics and lifestyles was collected by the electronic questionnaire, including sex, age, ethnicity, marital status, educational level, annual family income, residential location, drinking habit, smoking habit, dietary intake of cereal, fruits, red meat, vegetables, soybean products and aquatic products, and physical activity. The physical examination included height, weight, WC, systolic BP (SBP),

and diastolic BP (DBP). The WC was measured 1.0 cm above the navel and to the nearest 0.1 cm with light clothing. The SBP and DBP measurements were performed in a seated, upright position, and were instructed not to smoke, drink alcohol, coffee, or tea, or exercise for at least 30 minutes before the measurement. Both WC and BP were measured three times with a unified instrument, and then the mean values were taken. A series of biochemical tests were performed to obtain biological markers, and included cholesterol (CHOL), TG, HDL-C, total cholesterol (TC), FBG, serum uric acid (SUA, only in the CMEC, YBDS and HBDS), and glycated hemoglobin (HbA1c, only in the CMEC and YBDS). The venous blood samples were used to measure biochemical blood indexes at least 8 hours after fasting.

## Development of the MetS score

**Score development and assessment.** The confirmatory factor analysis (CFA), previously used to develop an MetS score in western countries on the basis of the continuous measures of the five MetS

components (WC, TG, HDL-C, SBP/DBP, and FBG)[7,14], was used in this study to develop the MetS score in a similar way. Some of the MetS components to be used were improved: SBP and DBP, with a high correlation which can result in a zero-matrix determinant and an ill-conditioned matrix that could not be inverted[31], were replaced by a mean arterial pressure (MAP) that has a better capability of predicting future MetS[32]; TG was log-transformed to reduce the skewness of distribution; HDL-C was inverted to make its higher value similar in interpretation to the other measures in the model. All variables of MetS components with different units were standardized for better integration. The CFA can estimate the factor loading (i.e., weight) of each MetS component, which indicates the degree of contribution of each component to the MetS score[33]. Two multi-group one-factor CFA were fit: Model 1 constrained the factor loadings to be equal across the eight combinations of age (young [<60 years] and old [≥60 years]), sex (male and female), and ethnicity (Han and minority); Model 2 allowed the factor loadings to vary across the eight subgroups, for developing age-sex-ethnicity-specific MetS scores. The differences in factor loadings across the subgroups were tested by the Chi-square test.

To evaluate the model's goodness of fit, several indexes of model fit were used, including Chi-square ($\chi^2$), Akaike's Information Criteria (AIC), Root Mean Square Error of Approximation (RMSEA), Standardized Root Mean Square Residual (SRMR), Comparative Fit Index (CFI), Goodness of Fit Index (GFI), McDonald's Fit Index (MFI), and Bentler-Bonett Normed Fit Index (NFI)[34]. A well-fitting model is expected to have as small $\chi^2$ and AIC as possible, with other indexes of fit meeting the recommended criteria: RMSEA < 0.08 (the smaller, the better); SRMR < 0.08 (the smaller, the better); CFI, GFI, MFI, and NFI ≥ 0.90 (the larger, the better)[35].

The MetS score of each individual was calculated based on the factor loading matrix and the specific variance matrix of the five MetS components, and then normalized (mean=0, standard deviation=1) for convenient comparison, with a higher MetS score representing more severe MetS. A linear model of the MetS score on the basis of original values of the MetS components was also fitted for each age-sex-ethnicity-specific group, allowing these equations to be applied to any individuals conveniently in a clinical context.

**Performance assessment of the MetS score.** The performance of the MetS score in predicting the incidence of the CVD-related risk factors and risk markers, selected according to the existing studies, was assessed in the CMEC baseline and follow-up datasets. The CVD-related risk factors used in this study included hypertension, diabetes, and dyslipidaemia[36], which were newly diagnosed in the medical examinations or blood tests at the CMEC baseline survey. Hypertension was defined as SBP ≥ 140 mmHg or DBP ≥ 90 mmHg, according to the International Society of Hypertension criteria (2020)[37]. Diabetes was defined as FBG ≥ 7.0 mmol/L or HbA1c ≥ 6.5%, according to the American Diabetes Association criteria (2019)[38]. Dyslipidaemia was defined as having one or more of the four conditions: TC ≥ 6.2 mmol/L (hypercholesterolemia), TG ≥ 2.3 mmol/L (hypertriglyceridaemia), HDL-C < 1.0 mmol/L (hypolipidaemia), and low-density lipoprotein cholesterol (LDL-C) ≥ 4.1 mmol/L (hyperlipidaemia), according to the Chinese Guidelines for Dyslipidaemia Management in Adults (2016)[39]. The CVD-related risk markers used in this study included HbA1c[40], CHOL[41], body mass index (BMI)[42], and SUA[43], with their abnormal status referred to as the elevated risk markers: (1) elevated HbA1c was defined as HbA1c > 6.5%;[44] (2) elevated CHOL was defined as CHOL > 6.2 mmo/L;[45] (3) elevated BMI was defined as BMI ≥ 28 kg/m²;[46] and (4) elevated SUA was defined as >417 μmol/L for men and >357 μmol/L for women[47].

Multiple logistic regression and linear regression were used to estimate the associations of the MetS score with the CVD-related risk factors and risk markers (i.e., the changes in the risk for these outcomes for each increase of 1 point in score) in the CMEC baseline,

respectively, after adjusting for a series of covariates (Table S1). The estimates were presented as odds ratios (ORs) and coefficients ($\beta$) with 95% confidence intervals (CIs) for multiple logistic and linear regression, respectively. Cox regression and generalized estimating equations (GEE) were used to estimate the associations of the MetS score with the CVD-related risk factors and risk markers (i.e., the changes in the rate of experiencing these outcomes for each increase of 1 point in score) in the CMEC follow-up, respectively, after adjusting for a series of covariates at baseline. In addition, the abnormal risk markers in the CMEC follow-up were also used as the outcomes to be associated with the MetS score by Cox regression. The estimates were presented as hazard ratios (HRs) and coefficients ($\beta$) with 95% CIs for Cox regression and GEE, respectively. The prediction performance of the MetS score in Cox regression was evaluated by the concordance index (C-index), which estimates the probability that the predicted outcome is consistent with the observed outcome. The C-index ranges from 0.5 (completely random) to 1 (completely consistent), with 0.5–0.7, ≥0.7–0.9, and ≥0.9 considered low, moderate, and high accuracy, respectively. To further assess the robustness of those associations in the study populations, all individuals were categorized into quartiles by their MetS scores, and the associations of the MetS score in quartiles with the CVD-related risk factors and the abnormal risk markers were estimated by Cox regression.

**Comparison with the traditionally defined MetS.** One was considered experiencing the traditionally defined MetS (a dichotomous variable) if presenting three or more of the five common MetS components, with each defined by a commonly used cut-off value: elevated WC was defined as ≥90 cm for men and ≥85 cm for women;[48] elevated TG were defined as a fasting TG level ≥1.7 mmol/L (150 mg/dL);[49] reduced HDL-C was defined as <1.03 mmol/L (< 40 mg/dL) in men or 1.29 mmol/L (< 50 mg/dL) in women;[49] elevated BP was defined as SBP ≥ 130 and/or DBP ≥ 85 mmHg; and elevated FBG was defined as ≥5.6 mmol/L[50]. The capacity of the MetS score in indicating traditionally defined MetS was evaluated by the Area Under the receiver operating characteristic (ROC) curve (AUC), the values of which range from 0.5 (no capacity) and 1.0 (complete capacity), with >0.7 and >0.9 considered acceptable and accurate, respectively. Also, the MetS score was converted into a dichotomous MetS variable to compare and evaluate the degree of consistency with the traditionally defined MetS. The maximum value of the Youden index, calculated as *sensitivity + specificity − 1*, estimated from the ROC curve was identified for the overall group and each age-sex-ethnicity-specific subgroup separately, to convert the MetS score into the dichotomous MetS for comparison with the traditionally defined MetS.

To demonstrate the clinical usefulness, the dichotomous MetS score, and the traditionally defined MetS were both used to predict the presence of ≥1 aforementioned risk factor (i.e., hyperlipidemia, diabetes, and hypertension), and the presence of ≥1 aforementioned risk marker (i.e., HbA1c, CHOL, BMI, and SUA). The same socio-demographic and lifestyle factors, as covariates, were adjusted for in the overall and age-sex-ethnicity-specific models. In the overall and each age-sex-ethnicity-specific subgroup, sensitivity and specificity were calculated and compared by the paired Chi-square test, and AUC was calculated and compared by Delong's test[51].

**Performance assessment of the MetS score in three external datasets**
The MetS score was calculated for each YBDS, HBDS, and FBDS participant by the equations established based on the CMEC baseline, and several analyses were conducted on the basis of the three datasets. First, the associations of the MetS score with the CVD-related risk factors and risk markers were estimated. Second, the participants in the three datasets were categorized into quartiles by the MetS score, and the associations of the MetS score in quartiles with the CVD-

related risk factors and risk markers were also estimated. Third, the maximum values of the Youden index estimated from the ROC curve, identified based on the CMEC baseline for the overall group and each age-sex-ethnicity-specific subgroup separately, were used to convert the MetS score into the dichotomous MetS variable, which was compared with the traditionally defined MetS by the ROC curve, assessed by AUC. Fourth, the dichotomous MetS score and the traditionally defined MetS were both used to predict the presence of ≥1 aforementioned risk factor (i.e., hyperlipidemia, diabetes, and hypertension), and the presence of ≥1 aforementioned risk marker (i.e., HbA1c, CHOL, BMI, and SUA), assessed by sensitivity, specificity, and AUC. A series of covariates were adjusted for (Tables S6, S10, S14), with age, sex, and ethnicity excluded from the covariates in the age-sex-ethnicity-specific models.

All statistical analyses were completed using R software version 4.0.2 (R Foundation for Statistical Computing), and statistical significance was declared if two-sided $P < 0.05$.

### Reporting summary
Further information on research design is available in the Nature Portfolio Reporting Summary linked to this article.

## Data availability
The datasets from this study are held in coded form, and legal data sharing agreements prohibit the authors from making the dataset publicly available. Access to individual deidentified participant data (including data dictionaries) may be granted to those who send a request with specific data needs, analysis plans, and dissemination plans to Prof. Peng Jia (e-mail: jiapengff@hotmail.com), and Shujuan Yang (e-mail: rekiny@126.com). The authors will give feedback within 30 days. However, individual identification information may not be available for public use.

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

## Acknowledgements

This study was supported by the National Natural Science Foundation of China (42271433, P.J.), Key R&D Project of Sichuan Province (2023YFS0251, S.Y.), the Fundamental Research Funds for the Central Universities (2042023kfyq04, P.J.), Remin Hospital of Wuhan University (JCRCYG-2022-003, P.J.), Regional Innovation Cooperation Program of Science and Technology Commission Foundation of Sichuan Province (2021YFQ0031, S.Y.), Chengdu Technological Innovation R&D Project (2021-YF05-00886-SN, S.Y.), Sichuan University-Dazhou Cooperation Project (2020CDDZ-26-SCU, S.Y.), Jiangxi Provincial 03 Special Foundation and 5G Program (20224ABC03A05, P.J.), Wuhan University Specific Fund for Major School-level Internationalization Initiatives (WHU-GJZDZX-PT07, P.J.), the Science and Technology Innovation Team of Hubei Province (P.J.), and the International Institute of Spatial Lifecourse Health (P.J.).

## Author contributions

S.Y. and P.J. designed the study; S.Y., B.Y., W.Y., S.D., C.F., and P.J. performed the data analysis; Y.S., X.Z., X.L., and T.H. prepared the data; S.Y., B.Y., and P.J. wrote the paper; S.Y., B.Y., W.Y., S.D., C.F., and P.J. provided comments and revision; S.D. developed an online tool; S.Y. and P.J. provided supervision.

## Competing interests

The authors declare no competing interests.
