## [Peer Review File · Nature Communications]

Development and validation of an age-sex-ethnicity-specific metabolic syndrome score in the Chinese adultsREVIEWER COMMENTS

Reviewer #1 (Remarks to the Author):

The authors have improved the manuscript but still things that need to be clarified before a final decision can be made.

Line 119, not clear what possible association 'reverse causality' could be applying to here. Do you mean that you did not use data on lipids, blood glucose or blood pressure from people who were taking medication to control these as their levels do not reflect the natural levels?

In line 380 and probably else where, the authors note the association between the MetS score and 'changes' in CVD risk markers. Do you mean differences in cross-sectional data? I don't remember seeing changes measured over time.

Para 386-96 describes previous continuous MetS scores developed in the US and Singapore yet notes only in western countries. Surely Singapore is not a western country even if quite wealthy. In this para western and developed are somehow mixed up.

Sentence lines 432-3 should say that 'central obesity in minorities may be attributed to low temperature, high altitude, high-fat food, and exercise-discouraging environments.'

Out of interest I looked at the online MetS Score calculator, the possible responses for characteristics and anthropometric indices seemed sensible but the options for biochemical measures didn't make sense if you were trying to enter actual measures. There were negative numbers, are these log-transformed? Why wouldn't you make it so the actual values are entered and log-transformation happens in calculating the score?

Line 440, remove 'ominous'.

How is the online calculator going to validate the MetS score?

Line 462 requires additional clarification. What is adjusted?

Reading the conclusion I do not understand the point about the MetS score helping to tailor therapy strategies. The score gives an idea of someone's overall risk but to select treatment strategies the doctor would have to know which aspects of the score were most important for an individual. I assume it would be possible for people to have similar scores but differing degrees of abnormality for each risk factor, and hence require different approach.

I have not checked references thoroughly but ref 1 is GBD which does not seem to fit with the refs at start of introduction.

Reviewer #2 (Remarks to the Author):

I appreciate the opportunity to review the authors' response letter. Unfortunately, I don't believe that the authors have addressed my initial concerns and questions adequately.

1. I completely agree with Reviewer #1's comment that the clinical implications of the MetS score are limited because patients are typically treated based on their individual levels of blood pressure, glucose, and lipid, rather than a single MetS score. The treatment for two individuals with the same MetS score could vary significantly depending on the specific components of MetS they exhibit. For example, one individual may have high blood pressure but normal blood glucose, while another individual may have normal blood pressure but high blood glucose. The treatment strategy for these two individuals would be quite different. Therefore, the MetS score cannot provide sufficient information for clinical decision-making and contains conceptual flaw.

2. The authors replied that a 50%-50% division of minorities and Hans is interpreted as a good, well-balanced generalisability. However, only about 5% of total population in Sichan, 6% in Chongqing, 33% in Yunnan, and 37% in Guizhou are minorities, which indicates an under-representation of Hans in the study population.

3. The authors replied that "genetically, Hans are highly similar across China regardless of region" to demonstrate the findings from Hans in minority regions can be applied to other parts of China. However, the authors described the rationale/hypothesis of their study in the second paragraph of Introduction "the components and progression of MetS may differ due to different genetic, demographic, socioeconomic, and lifestyle characteristics". That is to say, not only genetic, but also demographic, socioeconomic, and lifestyle characteristics affect MetS. This argument appears self-contradictory.

4. Although examining ethnic variations might primarily involve comparing Hans and other minorities, the study's objective is to develop a MetS score that accurately predicts CVD-related risk factors for both Hans and minorities. The differences within minorities cannot be ignored given that lifestyles, dietary patterns, and fat distributions vary substantially across the 55 minority ethnic groups.

Reviewer #1 (Remarks to the Author):

The authors have improved the manuscript but still things that need to be clarified before a final decision can be made.

Thanks for your efforts to re-evaluate and endorse our revision based on your great comments. We have addressed all of your remaining concerns below point by point.

Line 119, not clear what possible association 'reverse causality' could be applying to here. Do you mean that you did not use data on lipids, blood glucose or blood pressure from people who were taking medication to control these as their levels do not reflect the natural levels?

Yes, the reviewer was correct. We have used this suggestion to clarify that point (Lines 120-122).

In line 380 and probably else where, the authors note the association between the MetS score and 'changes' in CVD risk markers. Do you mean differences in cross-sectional data? I don't remember seeing changes measured over time.

Thanks for this question. In line 381, by saying "changes in CVD-related risk markers", we meant differences in CVD-related risk markers in the CMEC follow-up (Tables 4 and S5). We have clarified there by making two separate expressions (Lines 381-384).

Para 386-96 describes previous continuous MetS scores developed in the US and Singapore yet notes only in western countries. Surely Singapore is not a western country even if quite wealthy. In this para western and developed are somehow mixed up.

Thanks for catching this. We have revised it (Lines 388-389).

Sentence lines 432-3 should say that 'central obesity in minorities may be attributed to low temperature, high altitude, high-fat food, and exercise-discouraging environments.'

Thanks for catching this. We have revised it (Lines 434-436).

Out of interest I looked at the online MetS Score calculator, the possible responses for characteristics and anthropometric indices seemed sensible but the options for biochemical measures didn't make sense if you were trying to enter actual measures. There were negative numbers, are these log-transformed? Why wouldn't you make it so the actual values are entered and log-transformation happens in calculating the score?

Thanks for this question. It is the actual values that are entered in the online calculator. Instead of being log-transformed, the MetS scores from the calculator are z-

scores with both positive and negative values (Lines 180-181), which have also been used in previous studies developing such scores in western countries.

Line 440, remove 'ominous'.

Thanks for this suggestion. We have removed it (Line 443).

How is the online calculator going to validate the MetS score?

Thanks for this question. Initially we were thinking to have a written consent for all calculator users to share their data with us, so we can use the data to improve the MetS score continuously. However, there will be several stages to achieve this and we are just at the first stage. Therefore, to be more clear in this manuscript, we have removed all relevant words about using the calculator to validate the score (Lines 37-38, 89-90, 462-465).

Line 462 requires additional clarification. What is adjusted?

Thanks for this question. We meant that lifestyles and dietary patterns were adjusted in the models. We have clarified this (Lines 469-471).

Reading the conclusion I do not understand the point about the MetS score helping to tailor therapy strategies. The score gives an idea of someone's overall risk but to select treatment strategies the doctor would have to know which aspects of the score were most important for an individual. I assume it would be possible for people to have similar scores but differing degrees of abnormality for each risk factor, and hence require different approach.

Thank you for this comment. Only saying "tailor therapy strategies" was not clear (it was just general phrasing.), which actually meant tailoring CVD prevention strategies by age, sex, and ethnicity. But this point has been mentioned in the first sentence of the Conclusion section, so we have removed it here (Lines 489-490).

I have not checked references thoroughly but ref 1 is GBD which does not seem to fit with the refs at start of introduction.

Thanks for this suggestion. We have replaced the ref 1, and checked and fixed all issues in all references.

Reviewer #2 (Remarks to the Author):

I appreciate the opportunity to review the authors' response letter. Unfortunately, I don't believe that the authors have addressed my initial concerns and questions adequately.

Thanks for your great efforts to re-evaluating our revision and further explanations of your concerns. We have further addressed all of them or provided further explanations below point by point. We have also improved the limitations, so all potential issues could be realized by readers who could then use the MetS score and the calculator with caution.

1. I completely agree with Reviewer #1's comment that the clinical implications of the MetS score are limited because patients are typically treated based on their individual levels of blood pressure, glucose, and lipid, rather than a single MetS score. The treatment for two individuals with the same MetS score could vary significantly depending on the specific components of MetS they exhibit. For example, one individual may have high blood pressure but normal blood glucose, while another individual may have normal blood pressure but high blood glucose. The treatment strategy for these two individuals would be quite different. Therefore, the MetS score cannot provide sufficient information for clinical decision-making and contains conceptual flaw.

Thank you for these comments. What the reviewer mentioned above is raw information of MetS components, which is for sure among the most important information doctors need to know about patients for treatment. However, although patients could be treated according to levels of each MetS component, it is likely that levels of all components were close to the cut-off values but still on the normal side. In such cases, although individual components have no need to be treated (i.e., not meeting the dichotomous definition of MetS), their accumulation may lead to (severe) MetS.

MetS has been called 'an important clinical risk indicator' in many studies. The clinical usefulness of **an MetS score** in indicating the early risk for chronic diseases, including CVDs and diabetes, has been well justified and demonstrated in many studies. For example, the MetS score in western countries has been developed and widely used. In the article titled "An Examination of Sex and Racial/Ethnic Differences in the Metabolic Syndrome among Adults: A Confirmatory Factor Analysis and a Resulting Continuous Severity Score", published in the column "Clinical Science" of the journal *Metabolism* (<https://www.ncbi.nlm.nih.gov/pmc/articles/PMC4071942/>), the clinical importance of identifying different profiles of MetS components across demographics (e.g., sex, age, ethnicity) has been pointed out.

Also, in another article titled "Use of a Metabolic Syndrome Severity Z Score to Track Risk During Treatment of Prediabetes: An Analysis of the Diabetes Prevention Program", published in the column "Cardiovascular and metabolic risk" of the journal *Diabetes Care* (<https://pubmed.ncbi.nlm.nih.gov/30275282/>), it has been mentioned that this score can

accurately track reductions in risk during treatment and could be a compelling means of motivating patients, who may feel empowered by a decrease in score. This point has also been added to our justification. Therefore, we believe that the usefulness of an MetS score in clinical practice has been well justified in existing literature and also in this study.

The aforementioned explanations have been accepted by Reviewer #1. Hopefully they could suffice here as well.

2. The authors replied that a 50%-50% division of minorities and Hans is interpreted as a good, well-balanced generalisability. However, only about 5% of total population in Sichan, 6% in Chongqing, 33% in Yunnan, and 37% in Guizhou are minorities, which indicates an under-representation of Hans in the study population.

Thank you for this comment. Although the percentage of Hans is not proportional to the total population in the study region, our large sample size should suffice for our study aims. However, we have added this to the limitations (Lines 457-462).

3. The authors replied that "genetically, Hans are highly similar across China regardless of region" to demonstrate the findings from Hans in minority regions can be applied to other parts of China. However, the authors described the rationale/hypothesis of their study in the second paragraph of Introduction "the components and progression of MetS may differ due to different genetic, demographic, socioeconomic, and lifestyle characteristics". That is to say, not only genetic, but also demographic, socioeconomic, and lifestyle characteristics affect MetS. This argument appears self-contradictory.

Thanks for this comment. It was correct that demographic, socioeconomic, and lifestyle characteristics all affect MetS. We have developed the MetS score in sociodemographic groups. Lifestyle and dietary factors have been adjusted in our models, which has been made more clear in the limitations (Line 469-471).

4. Although examining ethnic variations might primarily involve comparing Hans and other minorities, the study's objective is to develop a MetS score that accurately predicts CVD-related risk factors for both Hans and minorities. The differences within minorities cannot be ignored given that lifestyles, dietary patterns, and fat distributions vary substantially across the 55 minority ethnic groups.

Thanks for this comment. Although we have mentioned the word "minorities" in the Introduction, it meant the minorities except Hans in all references cited. Therefore, we would not say examining heterogeneities across the 55 minority ethnic groups was our study aim, which has never been claimed in that way. However, we agree that the differences within minorities cannot be ignored, so we have added it to the limitation. We will make efforts towards it in our future work (Line 475-481).

REVIEWERS' COMMENTS:

Reviewer #1 (Remarks to the Author):

The authors have adequately addressed the issues.

Reviewer #2 (Remarks to the Author):

I appreciate the efforts made to revise the manuscript. However, my concerns haven't been adequately addressed; rather, they've been simply recognised as limitations. The aim of this paper is to develop an age-sex and ethnicity-specific MetS score for preventing CVD-related risk factors among Chinese individuals, not merely to explore the association between an established MetS score and CVD risk. Hence, the generalisability of this score is of paramount importance. Unfortunately, the study's Han population was chosen from China's minority regions (the Southwest), where only approximately 5% of Hans are living. The characteristics of MetS components, lifestyles, dietary habits, and fat distributions among these individuals vary significantly from those of the Hans living in the eastern half of the country. Therefore, the generalisability of the score is highly restricted. Additionally, the ethnicity-specific MetS score developed in this study is potentially misleading and unhelpful because the authors didn't account for the differences within different ethnic groups. Previous studies have highlighted significant differences in metabolic profiles across varying ethnic groups. For instance, the prevalence of diabetes is 6.5% among Tibetans, 6.3% among the Hui, but it's 11.5% in Uyghurs and 11.4% in Zhuangs.

Wang L, Gao P, Zhang M, Huang Z, Zhang D, Deng Q, Li Y, Zhao Z, Qin X, Jin D, Zhou M, Tang X, Hu Y, Wang L. Prevalence and Ethnic Pattern of Diabetes and Prediabetes in China in 2013. JAMA. 2017 Jun 27;317(24):2515-2523. doi: 10.1001/jama.2017.7596. PMID: 28655017; PMCID: PMC5815077.

Reviewer #3 (Remarks to the Author):

I agree with the concern of reviewers that MetS score has very limited clinical utility as well as utility in public health. The process of decision making and action taking through MetS

score based on the measured components of metabolic syndrome seem to be redundant and in reviewer's word "contains conceptual flaw". Clinical decision making on treatment choice or instruction on health promotion to improve the components used to calculate the score are based on concrete value, normal or abnormal , but not on the surrogate score. Also from individual perspective, they need to know where exactly went wrong in order to take action to improve.

I also share the concern regarding the generalizability of this score to other Han population living in more developed area in China because the sensitivity and specificity of the score might influenced by the prevalence of the conditions. I wonder if the authors could test this in other data sets with Han ethnic living in other area in China.

Reviewer #1 (Remarks to the Author):

The authors have adequately addressed the issues.

Thank you for recognizing the endorsing our revision efforts!

Reviewer #2 (Remarks to the Author):

I appreciate the efforts made to revise the manuscript. However, my concerns haven't been adequately addressed; rather, they've been simply recognised as limitations. The aim of this paper is to develop an age-sex and ethnicity-specific MetS score for preventing CVD-related risk factors among Chinese individuals, not merely to explore the association between an established MetS score and CVD risk. Hence, the generalisability of this score is of paramount importance. Unfortunately, the study's Han population was chosen from China's minority regions (the Southwest), where only approximately 5% of Hans are living. The characteristics of MetS components, lifestyles, dietary habits, and fat distributions among these individuals vary significantly from those of the Hans living in the eastern half of the country. Therefore, the generalisability of the score is highly restricted.

Thanks for this comment. We have applied this score to a large-sample provincially representative survey of Hans in eastern China. We have additionally applied it to another large-sample representative survey of Hans in central China. All results are robust, which has demonstrated the usefulness, and generalizability of this score. All new results have been added (**Tables S4, S5, S12-S17**). We have also made substantial revision in the main texts (e.g., **Lines 81-82, 101-115, 127-134, 137-152, 254-259, 381-396, 399-404**).

Additionally, the ethnicity-specific MetS score developed in this study is potentially misleading and unhelpful because the authors didn't account for the differences within different ethnic groups. Previous studies have highlighted significant differences in metabolic profiles across varying ethnic groups. For instance, the prevalence of diabetes is 6.5% among Tibetans, 6.3% among the Hui, but it's 11.5% in Uyghurs and 11.4% in Zhuangs.

Wang L, Gao P, Zhang M, Huang Z, Zhang D, Deng Q, Li Y, Zhao Z, Qin X, Jin D, Zhou M, Tang X, Hu Y, Wang L. Prevalence and Ethnic Pattern of Diabetes and Prediabetes in China in 2013. JAMA. 2017 Jun 27;317(24):2515-2523. doi: 10.1001/jama.2017.7596. PMID: 28655017; PMCID: PMC5815077.

Thanks for this comment. Although different metabolic profiles (e.g., prevalences of MetS components) across varying ethnic groups, what determines the MetS score are the weights of different MetS components in the composition of the score, which is not influenced by the prevalences of MetS components. We also found some inaccurate expressions in the Introduction, which have been revised (**Line 68**).

Reviewer #3 (Remarks to the Author):

I agree with the concern of reviewers that MetS score has very limited clinical utility as well as utility in public health. The process of decision making and action taking through MetS score based on the measured components of metabolic syndrome seem to be redundant and in reviewer's word "contains conceptual flaw". Clinical decision making on treatment choice or instruction on health promotion to improve the components used to calculate the score are based on concrete value, normal or abnormal, but not on the surrogate score. Also from individual perspective, they need to know where exactly went wrong in order to take action to improve.

Thank you for these comments. What was mentioned above is raw information of MetS components, which is for sure among the most important information doctors need to know about patients for treatment. However, although patients could be treated according to levels of each MetS component, it is likely that levels of all components were close to the cut-off values but still on the normal side. In such cases, although individual components have no need to be treated (i.e., not meeting the dichotomous definition of MetS), their accumulation may lead to (severe) MetS.

MetS has been called 'an important clinical risk indicator' in many studies. The clinical usefulness of **an MetS score** in indicating the early risk for chronic diseases, including CVDs and diabetes, has been well justified and demonstrated in many studies. For example, the MetS score in western countries has been developed and widely used. In the article titled "An Examination of Sex and Racial/Ethnic Differences in the Metabolic Syndrome among Adults: A Confirmatory Factor Analysis and a Resulting Continuous Severity Score", published in the column "Clinical Science" of the journal *Metabolism* (<https://www.ncbi.nlm.nih.gov/pmc/articles/PMC4071942/>), the clinical importance of identifying different profiles of MetS components across demographics (e.g., sex, age, ethnicity) has been pointed out.

Also, in another article titled "Use of a Metabolic Syndrome Severity Z Score to Track Risk During Treatment of Prediabetes: An Analysis of the Diabetes Prevention Program", published in the column "Cardiovascular and metabolic risk" of the journal *Diabetes Care* (<https://pubmed.ncbi.nlm.nih.gov/30275282/>), it has been mentioned that this score can accurately track reductions in risk during treatment and could be a compelling means of motivating patients, who may feel empowered by a decrease in score. This point has also been added to our justification. Therefore, we believe that the usefulness of an MetS score in clinical practice has been well justified in existing literature and also in this study.

The value of the MetS score has been increasingly recognized. For example, in a recent U.S. lifestyle intervention on cardiometabolic risk factors, published in an influential journal *Circulation* (<https://www.ncbi.nlm.nih.gov/pmc/articles/PMC7987882/>), the MetS score has been selected as an important outcome to assess the effectiveness of the trial.

From an individual perspective, although they need to know where exactly went wrong in order to take action to improve, previous studies have also considered the continuous MetS score a compelling means of motivating patients that may feel empowered by a decrease in score.

The aforementioned explanations have been accepted by both previous reviewers. Hopefully they could be understood here as well. To sum up, the MetS score was developed to improve, rather than to replace, the traditionally defined dichotomous MetS.

I also share the concern regarding the generalizability of this score to other Han population living in more developed area in China because the sensitivity and specificity of the score might influenced by the prevalence of the conditions. I wonder if the authors could test this in other data sets with Han ethnic living in other area in China.

Thanks for this comment. We have applied this score to a large-sample provincially representative survey of Hans in eastern China. We have additionally applied it to another large-sample representative survey of Hans in central China. All results are robust, which has demonstrated the usefulness, and generalizability of this score. All new results have been added (**Tables S4, S5, S12-S17**). We have also made substantial revision in the main texts (e.g., **Lines 81-82, 101-115, 127-134, 137-152, 254-259, 381-396, 399-404**).

REVIEWERS' COMMENTS

Reviewer #2 (Remarks to the Author):

Although the findings have been validated in other cohorts, I am still not positive about this paper. I concur with other reviewers that the MetS score developed in this paper has little clinical utility and potentially provides misleading information. The concept of combining different parameters into a single score is fundamentally problematic. The different components of MetS have distinct underlying causes, consequences, and treatments. For instance, the intervention and treatment needed for an individual with elevated blood glucose levels would differ substantially from an individual with abnormal lipid profile. However, the MetS score could be identical for both individuals, which could provide very misleading information for clinical decision-making.

Reviewer #2 (Remarks to the Author):

Although the findings have been validated in other cohorts, I am still not positive about this paper. I concur with other reviewers that the MetS score developed in this paper has little clinical utility and potentially provides misleading information. The concept of combining different parameters into a single score is fundamentally problematic. The different components of MetS have distinct underlying causes, consequences, and treatments. For instance, the intervention and treatment needed for an individual with elevated blood glucose levels would differ substantially from an individual with abnormal lipid profile. However, the MetS score could be identical for both individuals, which could provide very misleading information for clinical decision-making.

Thank you for endorsing that the metabolic syndrome (MetS) score developed in this study has been validated in other cohorts. The MetS score would be mainly used for early identification of CVD risks among the general or healthy population, rather than replacing individual MetS components for decision making in the choice of intervention and treatment strategies among those with abnormal MetS components. The raw information of MetS components is for sure among the most important information doctors need to know about patients for treatment. However, although patients could be treated according to levels of each MetS component, it is likely that levels of all components were close to the cut-off values but still on the normal side. In such cases, although individual components have no need to be treated (i.e., not meeting the dichotomous definition of MetS), their accumulation may lead to (severe) MetS.

MetS has been called 'an important clinical risk indicator' in many studies. The clinical usefulness of **an MetS score** in indicating the early risk for chronic diseases, including CVDs and diabetes, has been well justified and demonstrated in many studies. For example, the MetS score in western countries has been developed and widely used. In the article titled "An Examination of Sex and Racial/Ethnic Differences in the Metabolic Syndrome among Adults: A Confirmatory Factor Analysis and a Resulting Continuous Severity Score", published in the column "Clinical Science" of the journal *Metabolism* (<https://www.ncbi.nlm.nih.gov/pmc/articles/PMC4071942/>), the clinical importance of identifying different profiles of MetS components across demographics (e.g., sex, age, ethnicity) has been pointed out.

Also, in another article titled "Use of a Metabolic Syndrome Severity Z Score to Track Risk During Treatment of Prediabetes: An Analysis of the Diabetes Prevention Program", published in the column "Cardiovascular and metabolic risk" of the journal *Diabetes Care* (<https://pubmed.ncbi.nlm.nih.gov/30275282/>), it has been mentioned that this score can accurately track reductions in risk during treatment and could be a compelling means of motivating patients, who may feel empowered by a decrease in score. This point has also been added to our justification. Therefore, we believe that the usefulness of an MetS score in clinical practice has been well justified in existing literature and also in this study.

The value of the MetS score has been increasingly recognized. For example, in a recent U.S. lifestyle intervention on cardiometabolic risk factors, published in an influential journal *Circulation* (<https://www.ncbi.nlm.nih.gov/pmc/articles/PMC7987882/>), the MetS score has been selected as an important outcome to assess the effectiveness of the trial.

From an individual perspective, although they need to know where exactly went wrong in order to take action to improve, previous studies have also considered the continuous MetS score a compelling means of motivating patients that may feel empowered by a decrease in score.

The aforementioned explanations have been accepted by all other reviewers. Hopefully they could be understood here as well. To sum up, the MetS score was developed to improve, rather than to replace, the traditionally defined dichotomous MetS. We have added a paragraph to clarify this (**Lines 194-206**).